# The Role of Fluorescence In Situ Hybridization in the Surveillance of Non-Muscle Invasive Bladder Cancer: An Updated Systematic Review and Meta-Analysis

**DOI:** 10.3390/diagnostics12082005

**Published:** 2022-08-19

**Authors:** Weitao Zheng, Tianhai Lin, Zeyu Chen, Dehong Cao, Yige Bao, Peng Zhang, Lu Yang, Qiang Wei

**Affiliations:** Department of Urology, Institute of Urology, West China Hospital, Sichuan University, No. 37, Guoxue Road, Chengdu 610041, China

**Keywords:** fluorescence in situ hybridization, non-muscle invasive bladder cancer, surveillance, meta-analysis

## Abstract

Background: Fluorescence in situ hybridization (FISH) has become a popular biomarker for subsequent monitoring the recurrence of non-muscle invasive bladder cancer (NMIBC), several studies have investigated the ability of FISH to detect recurrence in the surveillance of NMIBC. However, the results were inconsistent. Methods: We conducted a systematic literature search extensively on authenticated databases including PubMed/Medline, Embase, Web of Science, Ovid, and Cochrane Library. Meta-analysis was performed to find out the sensitivity and specificity of FISH in predicting recurrence of NMIBC. Results: 15 studies were ultimately included in this meta-analysis, a total of 2941 FISH evaluations from 2385 NMIBC patients were available. The pooled sensitivity of FISH was 68% (95% CI: 0.58–0.76), and the pooled specificity was 64% (95% CI: 0.53–0.74). Subgroup analyses were performed in 7 studies without Bacillus Calmette–Guerin (BCG) treatment, the pooled sensitivity was 82% (95% CI: 0.68–0.90), and the pooled specificity was 63% (95% CI: 0.37–0.82). And in 9 studies using “UroVysion standard” to define positive FISH results showed a pooled sensitivity of 60% (95% CI: 0.50–0.70) and specificity of 70% (95% CI: 0.61–0.78). Conclusions: The findings of this study indicate that FISH has a satisfactory sensitivity (68%) and specificity (64%) and could be a potential biomarker in the surveillance of NMIBC. Moreover, BCG treatment and different FISH methods may have an impact on the sensitivity and specificity, these factors should be taken into account when making clinical strategy.

## 1. Introduction

Bladder cancer (BC) is the 11th most commonly diagnosed cancer worldwide with a significant risk of cancer morbidity and mortality. Approximately 75% of patients with BC present with a disease confined to the mucosa (stage Ta and carcinoma in situ) or submucosa (stage T1) which are defined as non-muscle invasive bladder cancer (NMIBC) [1,2]. Patients diagnosed with NMIBC need lifelong follow-up after transurethral resection of bladder tumor (TUR) based on the risk profile for the reason that NMIBC is at high risk of recurrence and progression [3]. The follow-up regimens consist of cystoscopy, urine cytology, and upper urinary tract imaging [1]. However, cystoscopy is invasive, relatively expensive and has the risk of morbidity concomitantly. Urine cytology is noninvasive and useful but its sensitivity for the detection of low-grade tumors is low. Computed tomography (CT) urography has poor sensitivity as well. Therefore, a noninvasive urinary biomarker with high sensitivity and specificity is needed to reduce the frequency of invasive testing in order to detect recurrent and progression in NMIBC patients and help predict response to therapies, which can improve quality of life and decrease costs [4,5]. Urine fluorescence in situ hybridization (FISH) is a technique to detect genetic alterations of chromosome 3, 7, 17 and the deletion of locus 9p21, which are most commonly associated with BC. FISH will not be influenced by the inflammatory response of the bladder to Bacillus Calmette–Guerin (BCG) compared to cystoscopy and urine cytology. Thus, it’s considered to be a potential biomarker for the surveillance of NMIBC patients undergoing intravesical instillation therapies [6,7,8]. Several studies have evaluated the ability of FISH to detect recurrence and predict progression. However, the results were inconsistent. Therefore, we conducted a systematic review and meta-analysis to synthesize available studies focused on FISH in surveillance of patients with NMIBC to assess its prognostic value for predicting recurrent disease.

## 2. Materials and Methods

### 2.1. Protocol for the Study

We conducted the meta-analysis referred to the protocol which had been registered in the International Prospective Register of Systematic Reviews database, and the protocol identification number was CRD42019121035 [9].

### 2.2. Evidence Acquisition and Selection Criteria

We conducted a systematic literature search extensively based on PubMed/Medline, Embase, Web of Science, Ovid and Cochrane Library. The searching strategy was used in the databases as cited: (“FISH” or “fluorescence in situ hybridization”) and (“bladder cancer” or “bladder carcinoma” or “urothelial carcinoma” or “urothelial cancer”) and (“recurrence” or “progression” or “surveillance” or “monitor”), Medical subject Headings (MeSH) words and free words were both used in the literature search to improve recall. The aim of the literature search was to identify studies focused on FISH in the surveillance of NMIBC published in English up to January 2021. After removing duplicates, a first screening was carried out based on the title and abstract. Studies qualified for further assessment must meet the following criteria: (1) randomized controlled or observational trials; (2) reported primary outcomes or available data to acquire sensitivity or specificity of FISH, as well as the control group; (3) containing at least 15 NMIBC patients. The exclusion criteria were as follows: (1) no full text available; (2) categorized as case, reports, editorials, reviews, letters, and meta; (3) containing less than 15 NMIBC patients.

Initially, we screened all included studies on titles, abstracts, irrelevant studies were excluded. The remaining studies were assessed by reading full-text articles exhaustively. Two independent reviewers participated in the screening process. When there were conflicts, the two reviewers discussed together until an agreement is reached, if failed to reach a consensus, a third senior reviewer was consulted to settle the claim.

### 2.3. Data Extraction and Quality Evaluation

Two independent reviewers conducted the data extraction process. The number of true positives, true negatives, false positives and false negatives were extracted from the included studies, as well as the characteristics of the patients such as author name, publish year, country, study design, age of the patients, the method and definition of positivity of FISH assay, follow-up time, definition of recurrence, postoperative adjuvant therapy, and parameters of correlated outcomes. Disputes were resolved by discussion or consultation with a third reviewer. The details were shown in Table 1. Standard quality evaluation of the included studies was performed based on the Newcastle–Ottawa scale and quality assessment of diagnostic accuracy studies-2 [10]. This study was conducted in line with the list of standards for reporting of diagnostic accuracy 2015 tools [11] and PRISAM 2009 checklist [12].

### 2.4. Statistical Analysis

The pooled sensitivity, specificity, positive predictive value (PPV), negative predictive value (NPV), and diagnostic likelihood ratios (DLRs) along with the 95% confidence intervals (CIs) were calculated using the extracted data. The sensitivity and specificity estimates were shown by forest plots using STATA 14.2 (StataCorp, College Station, TX, USA). The hierarchical summary receiver operating characteristic (HSROC) model and the bivariate model for meta analyses diagnostic accuracy test were carried out by metandi to summarize test performance command in STATA 14.2 (StataCorp) [13,14,15].These methods respected the binomial structure of diagnostic accuracy data and summarized paired measures synchronously such as sensitivity, specificity and positive or negative likelihood ratios. Furthermore, it was known that on account of different or implicit thresholds, heterogeneity was widespread across included studies. The bivariate/HSROC meta-analysis, a random effects approach, allowed pooling results [16]. Review Manager 5.3 (The Cochrane Collaboration, London, UK) was used to estimate the quality of the included studies.

## 3. Results

### 3.1. Study Selection and Identification

After comprehensive searching and rigorous evaluation, 35 studies were identified appropriate for initial assessment after screening titles and abstracts. Among the 35 relevant studies, 16 studies were excluded following the inclusion and exclusion criteria, and 19 studies were necessary for full-text articles assessed for eligibility. Then, 3 studies unable to extract data and 1 study fewer than 15 NMIBC patients were excluded. Finally, 15 studies with high reliability, adequate sample size, and comprehensible design with accessible data and full texts were included for ultimate meta-analysis (Figure 1).

### 3.2. Study and Patient Characteristics

These studies were published from 2005 to 2019, the number of NMIBC patients participated in these studies ranged from the highest 664 to the lowest 34 (Table 1). 9 of the 15 studies used the “UroVysion standard” to define positive cases according to the specifications of the kit [17], 5 studies used author defined standard for the discrimination of positive cases, that we called the “Author’s standard”. 1 study did not illustrate the positive definition. In the included studies, FISH results were collected at different times. Some studies performed FISH tests before and after BCG instillation, while some of them collected FISH results at 3 time points, including t0: before BCG; t1: at the end of BCG induction at 6 weeks; t2: 3 months after initial TUR.

### 3.3. Quality of Included Studies

We used Newcastle–Ottawa scale and QUADAS-2 tool to assess quality of the included studies. All included studies were scored above six (Table 1) based on the Newcastle–Ottawa scale and QUADAS-2 quality assessment result indicated low risk of bias (Figure 2). Hence, these studies were supposed to have a high rating of quality and very low risk of bias.

### 3.4. Overall Analysis

A total of 2941 FISH evaluations from 2385 NMIBC patients were available, 1078 (36.7%) of which developed recurrence events during the follow-up, including FISH positive in 700 (54.6%) and FISH negative in 378 (22.8%). The pooled sensitivity (Figure 3) of FISH in surveillance of NMIBC recurrence in the included studies was 68% (95% CI: 0.58–0.76), and the pooled specificity was 64% (95% CI: 0.53–0.74). The positive likelihood ratio (PLR) was 1.90 (95% CI: 1.44–2.51), the negative likelihood ratio (NLR) was 0.50 (95% CI: 0.38–0.65) (Figure 4). I^2^ test was performed to evaluate heterogeneity to select the appropriate calculation model. The heterogeneity was significant, I^2^ was 84.27% for sensitivity and 92.80% for specificity, respectively. Therefore, a random effect model was used. The summary receiver operating characteristic (SROC) plot could assess threshold effect, and heterogeneity of data in ROC space between sensitivity and specificity. The area under the curve (AUC) showed summary of test performance and the AUC in this study was 0.71 (95% CI: 0.36–0.92; Figure 5). The SROC curve was in the upper left corner, which demonstrated the best combination of sensitivity and specificity for the diagnostic test, so the diagnosis had a significant discriminatory potency.

### 3.5. Subgroup Analysis

Subgroup analysis was performed in 9 studies which used “UroVysion standard” to define positive FISH results and 7 studies whose patients did not receive BCG treatment after TUR. In “UroVysion standard” studies, the pooled sensitivity was 60% (95% CI: 0.50–0.70; Figure 6A), and the pooled specificity was 70% (95% CI: 0.61–0.78; Figure 6A). The AUC was 0.70 (95% CI: 0.66–0.74; Figure 6B). Among studies whose patients did not receive BCG treatment, the pooled sensitivity was 82% (95% CI: 0.68–0.90; Figure 6C), and the pooled specificity was 63% (95% CI: 0.37–0.82; Figure 6C). The AUC was 0.82 (95% CI: 0.78–0.85; Figure 6D).

## 4. Discussion

NMIBC patients have a high risk of recurrence and progression after transurethral resection, even though receiving adjuvant instillation therapy, and regular postoperative follow-up is essential to detect early recurrence in a timely manner [33].The combination of cystoscopy and urine cytology is considered the gold standard in the initial diagnosis of BC [34], and it is recommended for the surveillance of NMIBC [1]. Cystoscopy remains the primary modality for NMIBC postoperative surveillance. However, cystoscopy is invasive, painful and can make patients feel uncomfortable and dysphoric [18]. Long-term and frequent cystoscopies not only contribute to the physical pain and psychological burden of the patients, but also increase the risk of associated complications, particularly in old patients who are frail and have more underlying medical conditions. Furthermore, unnecessary cystoscopies without oncological benefits will increase the cost of the patients and waste the public healthcare resources [35,36]. Urine cytology is the most common used non-invasive urinary biomarker in the diagnosis and surveillance of NMIBC. Xie et al. conducted a meta-analysis to evaluate the diagnostic value of urine cytology in detecting BC, and the pooled sensitivity and specificity of urine cytology were 37% (95% CI, 0.35–0.39) and 95% (95% CI, 0.94–0.95), respectively [37]. Other studies reported the urine cytology with a sensitivity of 7.7–40.6% and a specificity of 88.3–98.0% [38,39,40,41]. Although cytology has a high specificity, the application is limited in terms of the low sensitivity [42]. It is essential to search for new non-invasive diagnostic biomarkers of both high sensitivity and specificity beyond cytology in predicting early recurrence of NMIBC [43].

FISH is non-invasive and convenient, previous studies have shown that FISH is a more accurate biomarker for detecting and predicting recurrence in NMIBC patients with a higher sensitivity than cytology. However, the specificity of cytology is superior when compared with FISH [44,45]. This is consistent with our results, the pooled sensitivity of FISH (68%) is much higher than that of cytology (7.7–40.6%) reported previously, although the pooled specificity (64%) of FISH is considerably high, there is still a gap from that of cytology (88.3–98.0%). Positive FISH results may help predict recurrence of NMIBC during the surveillance process, while the cystoscopy is negative but cytology is dubious [19,20,46].

Postoperative surveillance on the early recurrence of NMIBC is critical and relevant to the success of treatment, for the reason that early detection makes early intervention possible and avoids progression to MIBC to a certain extent. Once disease progressed to MIBC, patients usually need to receive radical cystectomy (RC) as a standard treatment, however, RC is associated with high risk of postoperative complications and reoperation [47]. Furthermore, the application of RC in elderly patients is controversial, particularly for those who are over 80 years old. Although extraperitoneal RC and ureterostomy have been indicated to have a limited mortality and mobility, the high risk of associated complications is still worrisome, especially in old patients with high level of frailty [48,49].

In addition to urine biomarkers, a number of clinical variables can also help predict the risk of recurrence and progression of NMIBC. Cicione et al. found that detrusor muscle thickness (DWT) measured by ultrasound was associated with NMIBC recurrence and progression, and patients with a greater DWT (>2.5 mm) had a higher risk of postoperative recurrence (OR 4.9, 95% CI: 2.5–9.5, *p* = 0.001) and progression (OR 2.21, 95% CI: 1.71–4.73, *p* = 0.001) [50]. FISH combined with DWT may be a non-invasive and available novel method with potentially clinically value to improve the accuracy of predicting recurrence and progression of NMIBC, which also can lengthen the surveillance intervals of cystoscopies and contribute to less pain and cost of the patients. Further studies are needed to explore and confirm the above assumption.

Liem et al. [51] conducted a meta-analysis in 2020 to assess the value of fluorescence in situ hybridization to predict early recurrence in patients with NMIBC at intermediate and high risk treated with BCG. 408 patients from 4 studies were included in the final analysis, recurrence hazard ratio (HR) was 1.20 (95% CI 0.81–1.79) before BCG, 2.23 (95% CI 1.31–3.62) at 6 weeks, 3.70 (95% CI 2.34–5.83) at 3 months and 23.44 (95% CI 5.26–104.49) at 6 months when FISH test is positive. Their result indicated that a positive FISH test result after BCG was associated with higher risk of recurrence, the preferred timing of FISH was 3 months after TUR. Liem’s study focused on the intermediate and high risk NMIBC patients treated with BCG, and reported HR in different time points, however, some studies including different stages of NMIBC patients, patients without BCG treatment after TUR and studies without time-related endings whose HR analysis was unavailable were not obtained in their analysis. As a result, the sample size was limited as well as the credibility and generalizability of the results. We conducted a systematic literature search extensively on authenticated databases and clinical trials published in English up to January 2022 focused on the utility of FISH in surveillance of patients with NMIBC were eligible. Our systematic review and meta-analysis included a total of 2941 FISH evaluations from 2385 NMIBC patients in initial 15 studies and reported the sensitivity and specificity as primary outcomes. Based on literature search, this meta-analysis is the largest and the latest study evaluating FISH focused on the utility of FISH in surveillance of patients with NMIBC to assess its prognostic value for predicting recurrence, which indicated that FISH has a satisfactory sensitivity and specificity in predicting recurrence in NMIBC patients.

Since BCG instillation is recommended for patients with intermediate- and high-risk tumors, the efficacy of FISH for the surveillance of NMIBC after BCG instillation is of great interest for practical purposes. Our results showed that the overall sensitivity of FISH for detection NMIBC recurrence was 68%, and the specificity was 64%, among studies whose patients did not receive BCG treatment, the pooled sensitivity was 82% and specificity was 63%, which indicated that BCG instillation may interfere with the FISH result. Possible explanations given for the lower accuracy for BCG instillation patients noted in the current study might be attributed to: (1) Follow-up time: longer follow-up time resulting in higher recurrence rate [21,52]. However, the follow-up time between BCG instillation studies and non-BCG instillation studies in our research has no statistical difference (95% CI: −14.19–8.00; *p* = 0.54). (2) BCG instillation therapy may have a screening effect, which can induce tumor mutations, and cause the difference between primary and recurrent urothelial carcinoma (UC). For recurrent UC, it may be necessary to detect the genes and chromosomes that may be mutated to find a more accurate urine marker [53]. Further investigation concerning different targeted genes or chromosomes may be needed to adjust the accuracy of FISH test for the surveillance of NMIBC patients undergoing BCG instillation.

**Table 1 diagnostics-12-02005-t001:** Characteristics of the included studies and patients.

Author	Year	Country	Study Design	Patients (No)	Mean Age (Years)	FISH Method	Time to FISH (no)	Follow-Up Time (mo)	NOS Score
Bollmann et al. [22]	2005	Germany	Retrospective	34	69.0	UroVysion FISH	NA	29.0	7
Mian et al. [23]	2006	Italy	Prospective	75	71.7	UroVysion FISH	NA	39.3	8
Mengual et al. [24]	2007	Spanish	Prospective	65	70.0	UroVysion FISH	Before the first BCG instillation	NA	7
Mengual et al. [24]	2007	Spanish	Prospective	65	70.0	UroVysion FISH	After the last BCG instillation	NA	7
Yoder et al. [19]	2007	USA	Prospective	250	72.0	UroVysion FISH	At the time of the cytologic diagnosis	23.0	8
Gofrit et al. [25]	2008	USA	Prospective	64	62.0	UroVysion FISH	At the conclusion of cystoscopy	13.5	8
Kipp et al. [26]	2009	USA	Retrospective	303	70.0	UroVysion probe set	NA	15.0	8
Savic et al.-1 [27]	2009	Switzerland	Prospective	68	71.0	Multitarget UroVysion	Before the first BCG instillation (*n* = 50) + before TUR (*n* = 18)	19.5	8
Savic et al.-2 [27]	2009	Switzerland	Prospective	68	71.0	Multitarget UroVysion	After the sixth BCG instillation.	19.5	8
Maffezzini et al. [28]	2010	Italy	Prospective	126	69.2	UroVysion FISH	NA	14.0	8
Galvan et al. [18]	2011	Spain	Prospective	222	72.9	Vysis FISH Pretreatment Reagent Kit	Before cystoscopy	8.0	7
Huang et al. [29]	2014	China	Prospective	41	NA	Bladder Cancer Kits	Before cystoscopy	29.5	8
Seideman et al. [20]	2015	USA	Retrospective	664	70.0	NA	NA	26.0	8
Todenhofer et al. [30]	2015	Germany	Prospective	114	70.0	UroVysion FISH	Before cystoscopy	24.0	7
Kamat et al. [21]	2016	USA	Prospective	95	66.0	UroVysion kit	At the first BCG instillation + 6 weeks + 3 months + 6 months	24.0	8
Liem et al. [31]	2017	The Netherlands	Prospective	114	70.7	UroVysion FISH	Before the first BCG instillation	23.0	8
Liem et al. [31]	2017	The Netherlands	Prospective	106	70.7	UroVysion FISH	At 6 weeks following TUR	23.0	8
Liem et al. [31]	2017	The Netherlands	Prospective	66	70.7	UroVysion FISH	At 3 months following TUR	23.0	8
Lotan et al. [32]	2019	USA	Prospective	150	72.0	UroVysion FISH	Before the first BCG instillation	9.0	7
Lotan et al. [32]	2019	USA	Prospective	133	72.0	UroVysion FISH	Before the sixth BCG instillation	9.0	7
Lotan et al. [32]	2019	USA	Prospective	118	72.0	UroVysion FISH	At 3 months after BCG initiation	9.0	7

FISH: Fluorescence in situ hybridization; TUR: transurethral resection of bladder tumor; BCG: Bacillus Calmette–Guerin; NA: not available; NOS: Newcastle–Ottawa.

It is worth mentioning that specificity of a few studies which did not use the UroVysion FISH method to test the specimen or follow the “UroVysion standard” to define a positive FISH result is 60%, lower than the pooled overall sensitivity of 68%. And the specificity is 70%, higher than the pooled overall specificity of 64%. Different FISH test methods seem to influence the sensitivity and specificity in some ways, so it is recommended that further researches focused on FISH perform the test according to a unified approach such as the UroVysion FISH method to avoid the bias and improve the accuracy of the results. Subgroup analyses revealed that BCG treatment and FISH methods may have an impact on FISH result.

Although the included studies have high quality, we acknowledge several limitations in this study. First, the definition of positive FISH result was not standardized with wide differences between studies which introduce heterogeneous results, but we performed subgroup analysis based on FISH methods, the results indicated that FISH test methods seem to influence the sensitivity and specificity, the more compliant FISH criteria could have led to overestimating positive FISH tests. Second, although we included 2941 FISH evaluations from 15 studies, the number of papers qualifying for the analysis is still small, therefore recall bias may have occurred. Third, HR analysis was not available since some studies didn’t have time-related endings. Our results should be interpreted with caution in consideration of the risk of bias across studies and statistical heterogeneity.

## 5. Conclusions

In this meta-analysis, our results reveal that FISH has a satisfactory sensitivity of 68% and specificity of 64% in predicting recurrence in patients with NMIBC, our findings suggest that FISH is a potential biomarker in the prognosis surveillance of NMIBC. Patients with positive FISH test still may need more intensive follow-up, even undertake precautionary therapies, such as additional cycles of maintenance instillation or conversion to radical cystectomy. However, BCG treatment and different FISH methods may have an impact on the sensitivity and specificity, these factors should be taken into account when making clinical strategy.

## Figures and Tables

**Figure 1 diagnostics-12-02005-f001:**
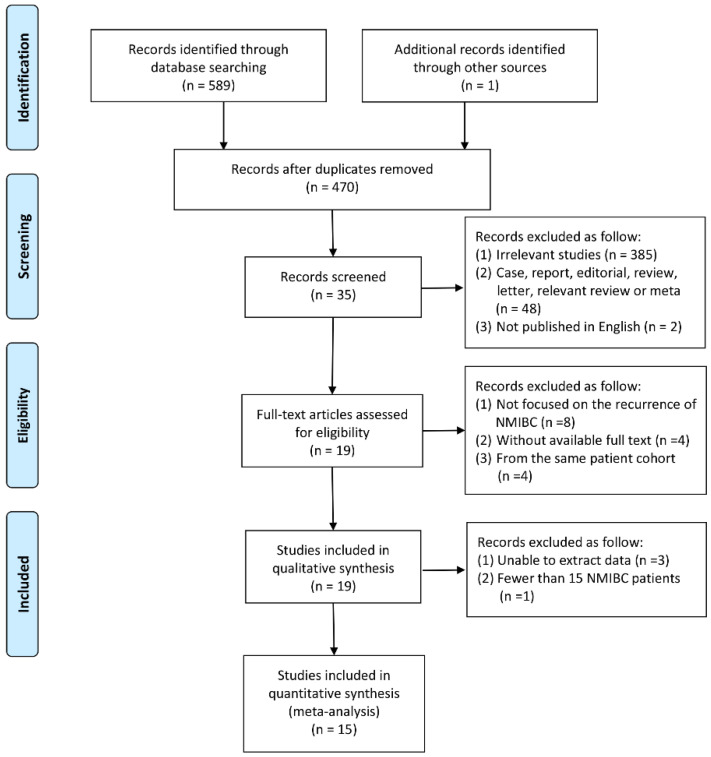
PRISMA 2009 flow diagram of the research.

**Figure 2 diagnostics-12-02005-f002:**
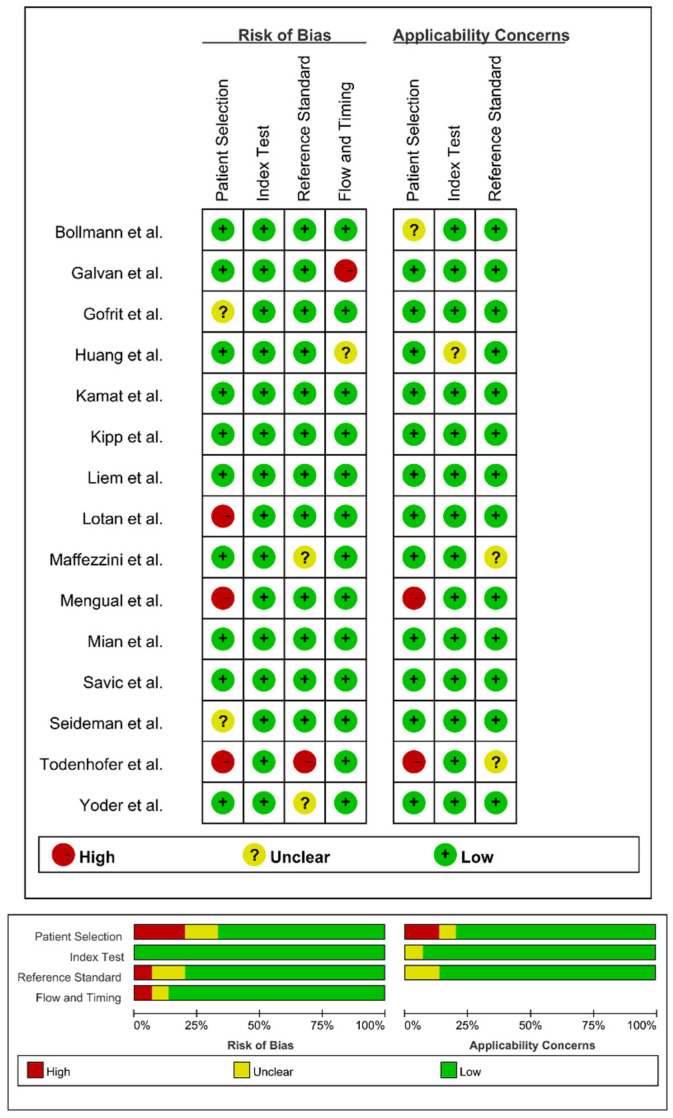
Overall quality assessment on the risk of bias and applicability concerns of the included studies using QUADAS-2 tool. Proportion of studies with high, unclear, and low risk of bias, and proportion of studies with high, unclear, and low concerns regarding applicability [18,19,20,21,22,23,24,25,26,27,28,29,30,31,32].

**Figure 3 diagnostics-12-02005-f003:**
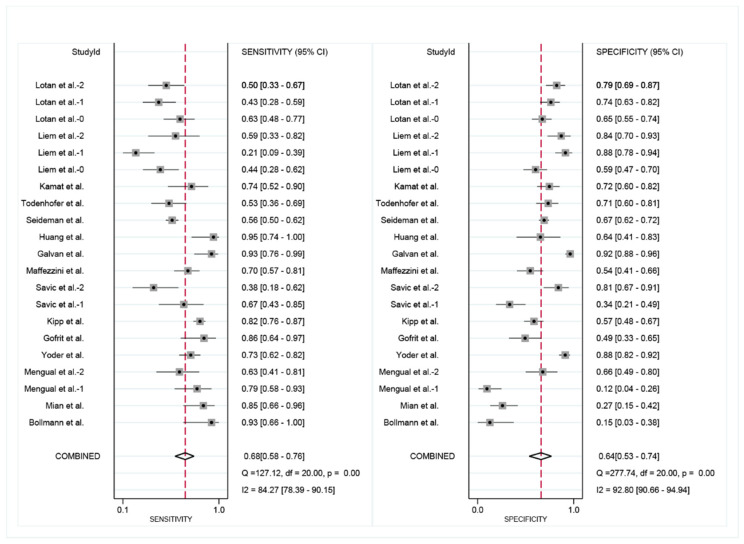
Paired forest plots of pooled sensitivity and specificity of FISH in surveillance of NMIBC recurrence. The point estimates of sensitivity and specificity from each study are indicate as squares and 95% confidence interval are shown with horizontal bars. (t2: Lotan et al.–2, Liem et al.–2, Mengual et al.–2; t1: Lotan et al.–1, Liem et al.–1, Savic et al.–2; t0: Lotan et al.–0, Liem et al.–0, Savic et al.–1, Mengual et al.–1) [18,19,20,21,22,23,24,25,26,27,28,29,30,31,32].

**Figure 4 diagnostics-12-02005-f004:**
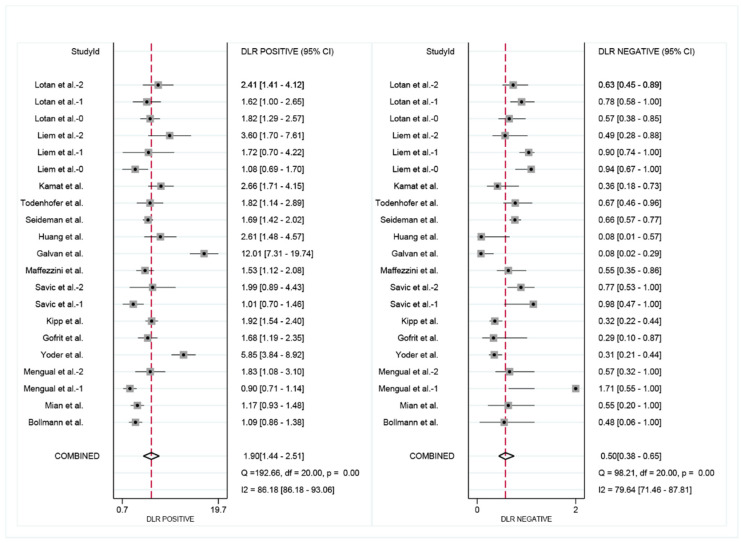
Positive likelihood ratio (PLR) and the negative likelihood ratio (NLR) of FISH in surveillance of NMIBC recurrence. (t2: Lotan et al.-2, Liem et al.-2, Mengual et al.-2; t1: Lotan et al.-1, Liem et al.-1, Savic et al.-2; t0: Lotan et al.-0, Liem et al.-0, Savic et al.-1, Mengual et al.-1) [18,19,20,21,22,23,24,25,26,27,28,29,30,31,32].

**Figure 5 diagnostics-12-02005-f005:**
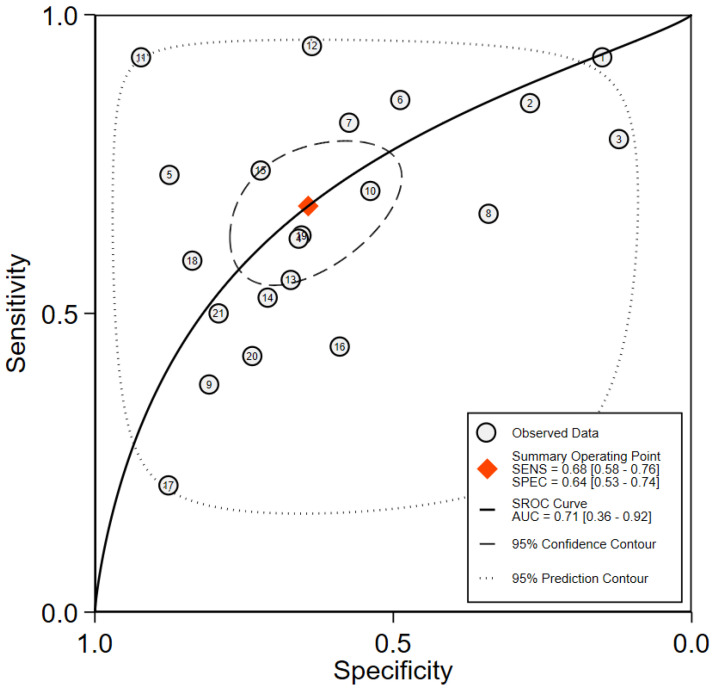
Summary receiver operating characteristic curve for the included studies of the research.

**Figure 6 diagnostics-12-02005-f006:**
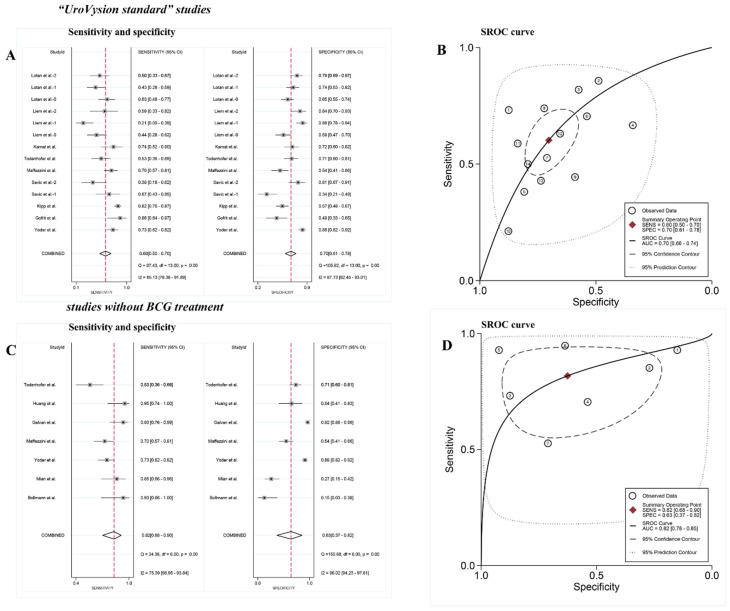
(**A**) Paired forest plots of pooled sensitivity and specificity of the “UroVysion standard” studies. (**B**) Summary receiver operating characteristic (SROC) plot of the “UroVysion standard” studies. (**C**) Paired forest plots of pooled sensitivity and specificity of the studies without BCG treatment. (**D**) Summary receiver operating characteristic (SROC) plot of the studies without BCG treatment [18,19,21,22,23,25,26,27,28,29,30,31,32].

## Data Availability

The data that support the findings of this study are available on request from the corresponding author Qiang Wei. The data are not publicly available due to them containing information that could compromise research participant privacy. We confirm that the authors are accountable for all aspects of the work (if applied, including full data access, the integrity of the data, and the accuracy of the data analysis) in ensuring that questions related to the accuracy or integrity of any part of the work are appropriately investigated and resolved.

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
