# Peer review of "The Role of Fluorescence In Situ Hybridization in the Surveillance of Non-Muscle Invasive Bladder Cancer: An Updated Systematic Review and Meta-Analysis"

_diagnostics, 2022, doi:10.3390/diagnostics12082005_

Round 1

Reviewer 1 Report

This is a meta-analysis about the usefulness of UroVysion FISH on urine specimens to detect early recurrence of non-muscle-invasive bladder cancer. This analysis included 15 studies with a total of 2941 FISH evaluations from 2385 NMIBC cases. So far it has been the largest study in this regard.

The analysis followed standard procedure of meta-analysis. Although the analysis itself has no serious flaws, there is one practical issue that the authors ought to discuss. UroVysion has not been widely accepted as a routine screening tool. One of the reasons is that the test is much more expensive and technically demanding than conventional cytology is. To claim the test is cost-effective, the authors must compare its sensitivity and specificity with those of conventional cytology. The authors mentioned a low sensitivity of cytology (33.4%), citing one single study (Ref 19) that is far from being representative. The authors should make more extensive literature review regarding urine cytology and compare the accuracies of two methods in the Discussion.

Reviewer 2 Report

The authors carried out a well-structured review on the use of FISH as biomarker for monitoring the recurrence of non-muscle invasive bladder cancer (NMIBC).

In my opinion, the review has been well carried out and reported by using  PRISMA criteria. Thus. I have just some minor suggestions aimed to improve your manuscript and due to the popularity of bladder cancer. All of them should be briefly cited in discussion to underline the importance of test like FISH for a very common disease as bladder cancer.

-        Some clinical variables have been evaluated with the aim to improve the calculation of progression and recurrence risk.  For instance, can you cite the following study where ultrasound detrusor wall was used to assess the risk of bladder cancer recurrence: A larger detrusor wall thickness increases the risk of non muscle invasive bladder cancer recurrence and progression: Result from a multicenter observational study- PMID: 29241311 DOI: 10.23736/S0393-2249.17.02992-7

-     In case of disease progression, radical cystectomy is often the primary choose of treatment however this is not devoid of complications. Furthermore, the disease progression occurs in older people and it is uneasy to identify patients at higher surgical risk. For this topic, can you cite the following studies where that issue has been evaluated :

Multicenter Analysis of Postoperative Complications in Octogenarians After Radical Cystectomy and Ureterocutaneostomy: The Role of the Frailty Index - PMID: 31402279 DOI: 10.1016/j.clgc.2019.07.002

Extraperitoneal radical cystectomy and ureterocutaneostomy in octogenarians - PMID: 21110093 DOI: 10.1007/s11255-010-9876-7
